# RMM: Reinforced Memory Management for Class-Incremental Learning

**Yaoyao Liu**[1]    **Bernt Schiele**[1]    **Qianru Sun**[2]

[1]Max Planck Institute for Informatics, Saarland Informatics Campus
[2]School of Computing and Information Systems, Singapore Management University
`{yaoyao.liu, schiele}@mpi-inf.mpg.de`   `qianrusun@smu.edu.sg`

## Abstract

Class-Incremental Learning (CIL) [40] trains classifiers under a strict memory budget: in each incremental phase, learning is done for new data, most of which is abandoned to free space for the next phase. The preserved data are exemplars used for replaying. However, existing methods use a static and ad hoc strategy for memory allocation, which is often sub-optimal. In this work, we propose a dynamic memory management strategy that is optimized for the incremental phases and different object classes. We call our method reinforced memory management (RMM), leveraging reinforcement learning. RMM training is not naturally compatible with CIL as the past, and future data are strictly non-accessible during the incremental phases. We solve this by training the policy function of RMM on pseudo CIL tasks, e.g., the tasks built on the data of the $0$-th phase, and then applying it to target tasks. RMM propagates two levels of actions: Level-1 determines how to split the memory between old and new classes, and Level-2 allocates memory for each specific class. In essence, it is an optimizable and general method for memory management that can be used in any replaying-based CIL method. For evaluation, we plug RMM into two top-performing baselines (LUCIR+AANets and POD+AANets [30]) and conduct experiments on three benchmarks (CIFAR-100, ImageNet-Subset, and ImageNet-Full). Our results show clear improvements, e.g., boosting POD+AANets by $3.6\%$, $4.4\%$, and $1.9\%$ in the 25-Phase settings of the above benchmarks, respectively. The code is available at `https://class-il.mpi-inf.mpg.de/rmm/`.

## 1   Introduction

Ideally, AI systems should be adaptive to ever-changing environments—where the data are continuously observed by sensors. Their models should be capable of learning new concepts from data while maintaining the ability to recognize previous ones. In practice, the systems often have constrained memory budgets because of which most of the historical data have to be abandoned [20]. However, deep-learning-based AI systems, when continuously updated using new data and limited historical data, often suffer from catastrophic forgetting, as the updates can override knowledge acquired from previous data [33, 34, 39].

To encourage research on the forgetting problem, Rebuffi et al. [40] defined a standard protocol of class-incremental learning (CIL) for image classification, where the training data of different object classes come in phases. In each phase, the classifier is evaluated on all classes observed so far. As the total memory size is limited [40], CIL systems abandon the majority of the data and only preserve a small number of exemplars, e.g., 20 exemplars per class, which will be used for replaying in subsequent phases. Replaying usually happens for multiple epochs [13, 18, 30, 40], so both the old

35th Conference on Neural Information Processing Systems (NeurIPS 2021).

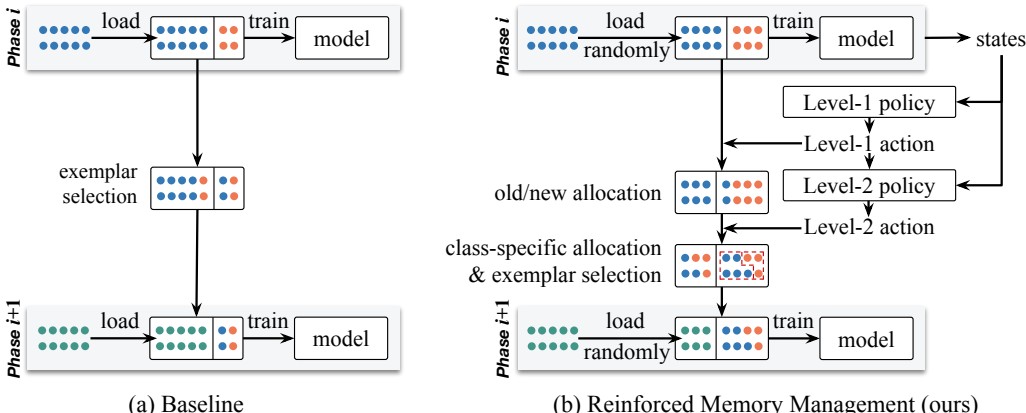

(a) Baseline                        (b) Reinforced Memory Management (ours)

Figure 1: (a) Existing CIL methods [18, 30, 40] allocate memory between old and new classes in an arbitrary and frozen way, causing the data imbalance between old and new classes and exacerbating the catastrophic forgetting of old knowledge in the learned model. (b) Our proposed method—Reinforced Memory Management (RMM)—is able to learn the optimal and class-specific memory sizes in different incremental phases. Please note we use orange, blue, and green dots to denote the samples observed in the $(i$-1)-th, $i$-th, and $(i$+1)-th phases, respectively.

class exemplars and new class data need to be stored in the limited memory. Existing CIL methods allocate memory between the old and new classes in an arbitrary and static fashion, e.g., 20 per old class *vs.* $1,300$ per new class for the ImageNet-Full dataset. This causes a serious imbalance between the old and new classes and can exacerbate the problem of catastrophic forgetting.

To address this, we propose to learn an optimal memory management policy for each incremental phase with continuously reinforced model performance and call our method reinforced memory management (RMM). Detailed actions include 1) allocating the memory between the existing (old) and the coming (new) data for each phase, and 2) specifying the memory for each old class according to its recognition difficulty before abandoning any of its data. To this end, we leverage reinforcement learning [26–28, 51, 59] and design a new policy function to contain two sub-functions that propagate two levels of actions in a hierarchical way. Level-1 function determines how to split memory between the old and new data. Its output action is then inputted into the Level-2 function to determine how to allocate memory for each old class. The overall objective of the function is to maximize the cumulative evaluation accuracy across all incremental phases. However, this is not naturally compatible with the standard protocol of CIL [40] where neither past nor future data are accessible for evaluation. To tackle this issue, we propose to pre-train the function on pseudo CIL tasks and then adopt it in the learning process of our target task. In principle, we can build such pseudo tasks using any available categorical data, e.g., the data in the 0-th phase of the target CIL task or the data from another dataset. Even though this is a non-stationary reinforcement learning problem, we can regard the pseudo and target CIL tasks as a sequence of stationary tasks and train the policy function to exploit the dependencies between these consecutive tasks. Such continuous adaptation in non-stationary environments is feasible based on the empirical analysis given in [2].

Technically, we propose the following method to guarantee the transferability of policy functions between pseudo and target CIL tasks. We take a Level-1 action based on the ratio of the number of new classes to the total number of classes observed so far. A lower (higher) ratio will result in weakening the stability (plasticity) of the classification model. Then, we take a Level-2 action for each individual class conditioned on both the Level-1 action and the training entropy of that class. A higher entropy denotes a more difficult class, leading to more memory allocated to the class. For evaluation, we conduct extensive CIL experiments by plugging RMM into two top-performing methods (LUCIR+AANets, POD+AANets) and testing them on three benchmarks (CIFAR-100, ImageNet-Subset, and ImageNet-Full). Our results show the clear and consistent superiority of RMM, e.g., it boosts the state-of-the-art POD+AANets by $3.6\%$, $4.4\%$, and $1.9\%$ in the 25-Phase settings of the above benchmarks, respectively.

Our technical contribution is three-fold. 1) A hierarchical reinforcement learning algorithm called RMM to manage the memory in a way that can be conveniently modified through incremental phases

and for different classes. 2) A pseudo task generation strategy that requires only in-domain available data (small-scale) or cross-domain datasets (large-scale), relieving the data incompatibility between reinforcement learning and class-incremental learning. 3) Extensive experiments, visualization, and interpretation for RMM in three CIL benchmarks and using two top models as baselines.

## 2   Related Work

**Incremental Learning** [22, 36, 48, 53, 58] continuously updates the model using data coming in a sequence of phases. Similar tasks are also referred to as continual learning [12, 32] and lifelong learning [3, 9]. Recent papers are either task-incremental learning—each phase corresponds to a task (dataset) that contains new data of all seen classes [7, 10, 19, 29, 43, 47, 57], or class-incremental learning (CIL)—each phase contains data of a new set of classes, i.e., classes are unseen [4, 6, 18, 25, 31, 37, 40, 41, 49, 52, 55–57]. This paper is concerned with CIL. The key challenge of CIL is the forgetting problem—older classes are forgotten in later phases. Existing methods tackling this can be divided into three categories: memory-based, regularization-based, and network-architecture-based [11, 35]. *Memory-based* methods preserved a small subset of the old class data (exemplars) to replay the model on them (together with the new class data), in order to relieve the forgetting of the old classes. Some work [18, 40] proposed heuristic strategies to select more representative exemplars from the old class data, and others [31, 47] tried to generate exemplars in optimizable frameworks. None of them changed the allocation of memory for different classes, i.e., all used an arbitrary and static scheme for memory allocation. *Regularization-based* methods introduce regularization terms in the loss function to consolidate previous knowledge when training the model on new data. The key idea is to enforce predicted label logits [29, 40], features maps [13, 18], or the topology in the feature space [49] of the new model to be close to that of the previous model. *Network-architecture-based* methods aim to design "incremental network architectures". Some work [45, 54] gradually extended the network capacity for new data, while others proposed to freeze partial network parameters [1, 30] to preserve the knowledge of the old classes.

**Reinforcement Learning** defines an agent that needs to decide its actions in an unknown environment by maximizing the expected cumulative reward. It has been widely applied to many optimization problems, e.g., neural architecture search [54, 59] and neural machine translation [38, 46]. Reinforcement learning has also been introduced to solve incremental learning problems. Xu et al. [54] proposed to increase convolution filters once a new task arrives and optimize the increased number by reinforcement learning. Gao et al. [14] proposed an improved version that makes the minimal expansion of the network, reducing memory and computing overheads. Veniat et al. [50] introduced a modular architecture, where each module represents a different atomic skill, and used the RE-INFORCE algorithm [51] to optimize it. Huang et al. [21] combined reinforcement learning with Net2Net [8] and designed a NAS-based CIL method. In our work, we also use the REINFORCE algorithm [51], but **differ** in three aspects. First, we are the first to optimize memory allocation for CIL in a reinforced way. Second, we learn the policy functions on generated pseudo CIL tasks, where we can access both past, and future data (for each incremental phase) and thus are able to compute the cross-phase (long-term) rewards. In contrast, the related work [14, 54] could use only current-phase data to estimate a short-term reward. Third, our reinforcement learning has a hierarchical structure that specially fits the nature of the data stream in the CIL settings.

## 3   Preliminaries

**Class-Incremental Learning (CIL)** usually assumes $(N+1)$ learning phases: an initial phase and $N$ incremental phases during which the number of classes gradually increases till the maximum [13, 18, 20, 31]. We assume that total memory $\mathcal{M}$ is bounded and fixed for all incremental phases [40]. $\mathcal{M}$ is used to store the exemplars and new coming data as both kinds of data need to be loaded repeatedly during training epochs. In the initial (0-th) phase, data $\mathcal{D}_0$, containing the training samples of $\mathcal{C}_0$ classes, are used to learn the initial classification model $\Theta_0$. In the $i$-th incremental phase, we split $\mathcal{M}$ into two dynamic partitions: the exemplar memory $\mathcal{M}_{\text{old}}$ and new data memory $\mathcal{M}_{\text{new}}$. We select $\mathcal{E}_t$ as representative samples of the data seen in the $t$-th phase, and denote total exemplars $\mathcal{E}_0 \sim \mathcal{E}_{i-1}$ shortly as $\mathcal{E}_{0:i-1}$. We save $\mathcal{E}_{0:i-1}$ into $\mathcal{M}_{\text{old}}$ and free $\mathcal{M}_{\text{new}}$. Then, we observe new data that contain $\mathcal{C}_i$ new classes. We randomly load new data into $\mathcal{M}_{\text{new}}$ until $\mathcal{M}_{\text{new}}$ is full, and all the other new data

are discarded. We denote the loaded new data as $\mathcal{D}_i$. Then, we initialize $\Theta_i$ with $\Theta_{i-1}$, and train it using $\mathcal{E}_{0:i-1} \cup \mathcal{D}_i$. The resulting model $\Theta_i$ will be evaluated with a test set containing all classes observed so far. We repeat this training and testing, and report the average accuracy across all phases.

**Reinforcement Learning (RL)** aims to learn an optimal policy function $\pi$ for an agent interacting in an unknown environment [51, 54, 59]. In the CIL scenario, in each incremental phase, the agent observes the current state $s_i$ from the environment, and then takes an action $a_i$ (how to allocate memory) according to the policy function $\pi(a_i|s_i)$. Subsequently, the environment is updated to a new state $s_{i+1}$ and the reward $r_i$ is calculated to optimize the parameters of $\pi(a_i|s_i)$ through back-propagation. Specifically, the learning objective of $\pi(a_i|s_i)$ is to maximize the expected cumulative reward $R_i = \sum_{t=i}^{\infty} \gamma^{t-i} r_t$, where $\gamma \in [0,1)$ is a discounting factor that determines the weights of future rewards. Please note that in our case, the $(N+1)$-phase CIL task is a finite horizon problem [15, 59], so we remove the discounting factor and use $R = \sum_{t=0}^{N} r_t$, which is actually the cumulative validation accuracy of all training CIL tasks. In Section 4, we discuss the proposed RL algorithm for memory allocation and how to generate pseudo tasks for training its policy function.

# 4 Reinforced Memory Management (RMM)

Our RMM approach learns policy functions that propagate two levels of actions in a hierarchical way, specially designed for CIL. As illustrated in Figure 1 (b), Level-1 determines the memory split between exemplars and new data, and Level-2 allocates the memory for each individual class. We motivate and introduce the formulation of RMM, including the definitions of states, actions, rewards, and hierarchical policy functions in Section 4.1. In Section 4.2, we detail the steps of creating pseudo CIL tasks on which we learn the policy functions. In Section 4.3, we summarize the algorithm.

## 4.1 Formulation

In the $i$-th incremental phase CIL, we manage the memory for two kinds of data: exemplars $\mathcal{E}_{0:i-1}$ and new data $\mathcal{D}_i$. For the former, we have access to their images and labels so we can allocate a different memory size to a different class, e.g., based on its recognition difficulty. For the latter, we do not have such access before loading the data (otherwise, causing a violation to the CIL protocol), so we are only able to learn a total memory size, i.e., the memory size for all new classes (and then split it evenly for each individual class). Therefore, the memory management in CIL settings is inherently hierarchical: 1) coarse memory allocation between exemplars and new data; and then 2) fine-grained memory allocation among specific classes. To this end, we modify the standard reinforcement learning into a hierarchical structure.

As illustrated in Figure 2 (a), in the $i$-th incremental phase of CIL (i.e., the environment), the argent receives a state value $s_i$. Level-1 policy $\pi_\eta$ takes $s_i$ as the input to produce an action $a_i^{[1]} \sim \pi_\eta(s_i)$. $a_i^{[1]}$ determines how to split memory between the exemplars and new data. After that, Level-2 policy $\pi_\phi$ takes $s_i$ and $a_i^{[1]}$ as inputs to produce the second action $a_i^{[2]} \sim \pi_\phi(s_i, a_i^{[1]})$ that distributes the exemplar memory for each individual class.

**States**, defined for our CIL settings, should have two properties. 1) Being transferable between CIL tasks, e.g., from a small-scale CIL task including 50 classes (in total) to a large one including 100 classes. The reason is that we need to transfer the policy functions learned from pseudo CIL tasks (defined in Section 4.2) to the target task. The states, the inputs of policy functions, should also be transferable. 2) Being distinct in each incremental phase. This is to enable the state variable to represent a specific forgetting or data imbalance degree at each different learning phase of the CIL model. To fulfill these properties, we formulate the state in the $i$-th phase as $s_i = \left( \frac{\mathcal{C}_i}{\sum_{t=0}^{i-1} \mathcal{C}_t}, \frac{|\mathcal{M}_{\text{old}}|}{|\mathcal{M}|} \right)$, where $\mathcal{C}_i$ denotes the number of classes in $\mathcal{D}_i$, $\mathcal{M}_{\text{old}}$ denotes the memory allocated to exemplars $\mathcal{E}_{0:i-1}$, and $\mathcal{M}$ is the total memory.

**Level-1 Actions.** In the 1-st incremental phase, our Level-1 policy function produces an action to allocate the memory for exemplars $\mathcal{E}_0$ and new data $\mathcal{D}_1$. We denote this action as $a_1^{[1]}$ and assign its value with the ratio of the number of the exemplars $|\mathcal{E}_0|$ to the memory size $|\mathcal{M}|$, so we have $a_1^{[1]} \in (0,1)$. In the $i$-th phase ($i \geq 2$), the definition of $a_i^{[1]}$ is different to $a_1^{[1]}$ as it is a relative change over $a_{i-1}^{[1]}$. Specifically, $a_i^{[1]}$ is the ratio of increased (if its value is positive) or decreased (if

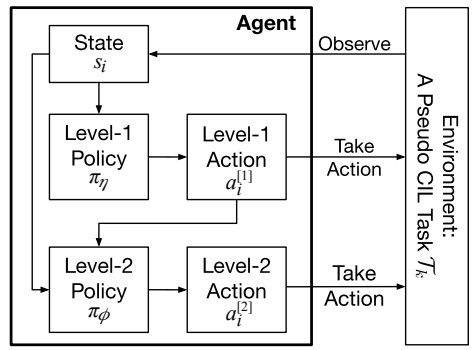
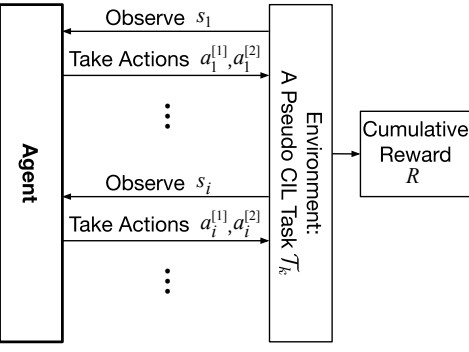

(a) The $i$-th incremental phase
of the $k$-th pseudo CIL task

(b) The $k$-th pseudo CIL task

Figure 2: (a) In the $i$-th phase of the $k$-th pseudo CIL task, Level-1 policy $\pi_\eta$ takes $s_i$ as the input, and produces action $a_i^{[1]}$. Level-2 policy $\pi_\phi$ takes $s_i$ and $a_i^{[1]}$ as the inputs, then produces action $a_i^{[2]}$. (b) For the $k$-th pseudo CIL task, we allocate memory for $N$ times (i.e., in $N$ phases) using the policies $\pi_\eta$ and $\pi_\phi$, and compute the cumulative reward $R$.

negative) memory size of $\mathcal{M}_{\text{old}}$ compared to the $(i\text{-}1)$-th phase. Using this definition aims for smooth and continuous memory management. In the formulation, the memory sizes of exemplars $\mathcal{E}_{0:i-1}$ and new data $\mathcal{D}_i$ are, respectively,

$$|\mathcal{M}_{\text{old}}| = |\mathcal{E}_{0:i-1}| = \sum_{t=1}^{i} a_t^{[1]}|\mathcal{M}|, \quad |\mathcal{M}_{\text{new}}| = |\mathcal{D}_i| = \left(1 - \sum_{t=1}^{i} a_t^{[1]}\right)|\mathcal{M}|. \quad (1)$$

We set a constrain $a_i^{[1]} \in [-0.1, 0.1]$ for $i \geq 2$. Otherwise, if $a_i^{[1]}$ is too big, there are not enough exemplars to fill the memory, as most old-class data has been abandoned. If $a_i^{[1]}$ is too small, many exemplars will be permanently deleted in this phase, making it hard or even impossible to adjust $\mathcal{M}_{\text{old}}$ back to a high value in the future phases. If $\sum_{t=1}^{i} a_t^{[1]} > 1$, $\mathcal{M}_{\text{new}}$ will be negative. So, we force $\sum_{t=1}^{i} a_t^{[1]} \leq 1$ by rejection sampling [5], i.e., using $\pi_\eta$ to output another action until it is feasible to execute. Note that this situation rarely happens in real training, because when $\mathcal{M}_{\text{new}}$ becomes very low, $\pi_\eta$ tends to produce an action to increase it.

**Level-2 Actions.** Here, we elaborate on how to get class-specific memory allocation. In the $(i-1)$-th phase, we split the classes for $\mathcal{D}_{i-1}$ into two groups evenly according to training entropy values: classes with higher values (difficult classes) are in one group and the rest in the other group. Therefore, Level-2 action $a_i^{[2]} \in (0, 1)$ determines how to split memories between harder and easier classes. During initial experiments, we observed that using two groups already yields improved results and using more groups causes a decrease.

Let $\mathcal{M}_j^A$ and $\mathcal{M}_j^B$ denote the memory allocated for the high-entropy and low-entropy groups, respectively, in the $j$-th phase ($j \leq i$):

$$|\mathcal{M}_j^A| = a_{j+1}^{[2]}|\mathcal{E}_j| = \frac{a_{j+1}^{[2]}\mathcal{C}_j}{\sum_{t=1}^{i}\mathcal{C}_t}|\mathcal{M}_{\text{old}}|, \quad |\mathcal{M}_j^B| = (1 - a_{j+1}^{[2]})|\mathcal{E}_j| = \frac{(1 - a_{j+1}^{[2]})\mathcal{C}_j}{\sum_{t=1}^{i}\mathcal{C}_t}|\mathcal{M}_{\text{old}}|. \quad (2)$$

Then, we allocate memory evenly to the classes within the group, e.g., if the high-entropy group has 10 classes, each class will have a memory size of $\frac{1}{10}|\mathcal{M}_j^A|$.

**Rewards.** The objective of CIL is that the trained model (in any phase) should be efficient to recognize all classes seen so far. It is intuitive and convenient to use the validation accuracy as the reward in each phase. In the $i$-th phase, the objective of RMM is to maximize the expected cumulative reward, i.e., $R = \sum_{i=0}^{N} r_i$, where $r_i$ denotes the validation accuracy in the $i$-th phase.

## 4.2 Optimization

In the CIL protocol, it is impossible to see past or future data in any incremental phase. It is thus not intuitive how to compute cumulative rewards till the last phase. We propose to solve the issue by generating pseudo CIL tasks (where all data are accessible).

**Pseudo CIL Tasks** should meet two requirements: 1) their training and validation data are fully accessible for computing cumulative rewards, and 2) they have the same format (e.g., the same number of phases) of the target CIL task. ***Data Sources:*** For requirement 1, an intuitive solution is to use $\mathcal{D}_0$ (available in the 0-th phase). Based on the CIL protocol [13, 18, 20, 30], $\mathcal{D}_0$ contains half of the classes of the whole dataset, e.g., 50 classes on CIFAR-100, which supplies enough data to build downsized CIL tasks. When building the tasks, we randomly choose 10% training samples of each

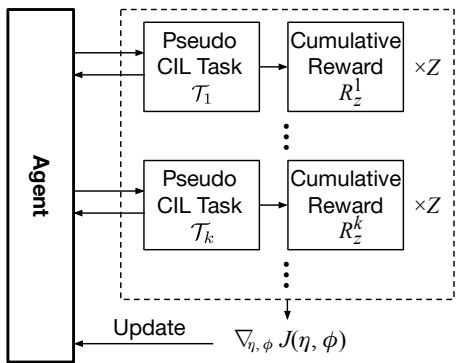

Figure 3: Updating $\eta$ and $\phi$ in one epoch. To get stable gradients for $J(\eta, \phi)$, we create $K$ different pseudo CIL tasks, and run each task for $Z$ times.

class (from $\mathcal{D}_0$) to compose a pseudo validation set (note that we are not allowed to use the original validation set in training). When aiming for larger-scale data in CIL, we can leverage smaller datasets. For example, the pseudo tasks for ImageNet-Subset can be built on the data of CIFAR-100. This is also meaningful to evaluate the transferability of RMM policy functions (discussed in the Ablation Study). ***Task Generation Protocol*** is based on requirement 2. If using another dataset, we simply follow its original CIL protocol. If using the data accessed in the 0-th phase (i.e., $\mathcal{D}_0$), we can reduce the number of classes (in each phase) by half. For example, for CIFAR-100, we use 50-class $\mathcal{D}_0$ to generate a 5-phase pseudo CIL task as follows: loading 25 classes in the 0-th phase, and after that, five classes per phase. To generate another pseudo task, we simply change the order of classes.

**Training.** We elaborate the steps of learning Level-1 policy $\pi_\eta$ and Level-2 policy $\pi_\phi$ in the following. The goal is to optimize the parameters $\eta$ and $\phi$ by maximizing the expected cumulative reward $J(\eta, \phi)$. We denote any pseudo CIL task and its cumulative reward as $\mathcal{T}$ and $R$, respectively, and have,

$$J(\eta, \phi) = \mathbb{E}_{\mathcal{T}} \mathbb{E}_{\pi_\eta, \pi_\phi}[R]. \tag{3}$$

*Policy Gradient Estimation*. According to the policy gradient theorem [51], we can compute the gradients for $J(\eta, \phi)$ as follows,

$$\nabla_{\eta, \phi} J(\eta, \phi) = \mathbb{E}_{\mathcal{T}} \left[ \sum_{i=1}^{N} \mathbb{E}_{\pi_\eta, \pi_\phi} [\nabla_{\eta, \phi} \log(\pi_\eta(a_i^{[1]}|s_i) \pi_\phi(a_i^{[2]}|s_i, a_i^{[1]})) R] \right]. \tag{4}$$

Following the REINFORCE algorithm [51], we replace the expectations $\mathbb{E}_{\mathcal{T}}[\cdot]$ and $\mathbb{E}_{\pi_\eta, \pi_\phi}[\cdot]$ with sample averages using the Monte Carlo method [16]. Specifically, in each epoch, we create $K$ pseudo tasks and run each task for $Z$ times, as shown in Figure 3. Thus we can derive the empirical approximation of $\nabla_{\eta, \phi} J(\eta, \phi)$ as,

$$\nabla_{\eta, \phi} J(\eta, \phi) = \frac{1}{ZK} \sum_{k=1}^{K} \sum_{z=1}^{Z} \sum_{i=1}^{N} \nabla_{\eta, \phi} \log(\pi_\eta(a_i^{[1]}|s_i) \pi_\phi(a_i^{[2]}|s_i, a_i^{[1]}))(R_z^k - b), \tag{5}$$

where $R_z^k$ denotes the $z$-th reward for the $k$-th pseudo task $\mathcal{T}_k$, and $b$ denotes the baseline function—the moving average of previous rewards. Using this baseline function is a common trick in RL to reduce the variance of estimated policy gradients [23, 42, 59].

***Updating Parameters***. We update $\eta$ and $\phi$ in each epoch according to the gradient ascent rule [54, 59]:

$$\eta := \eta + \beta_1 \nabla_\eta J(\eta, \phi), \quad \phi := \phi + \beta_2 \nabla_\phi J(\eta, \phi), \tag{6}$$

where $\beta_1$ and $\beta_2$ are the learning rates. We iterate this update for $m$ epochs in total.

### 4.3 Algorithm

Algorithm 1 summarizes the overall training steps of the proposed RMM. There are four loops in the algorithm: 1) we train the RMM agent for $m$ epochs; 2) we create $K$ pseudo CIL tasks in each epoch; 3) we run each pseudo CIL task for $Z$ times; and 4) there are $N$+1 learning phases each time. Specifically, Line 3 initializes the parameters of policy functions. Line 6 creates the $k$-th pseudo CIL task. Line 8 initializes the classification model. Lines 10-16 allocate the memory according to the actions produced by RMM policy. Line 17 loads new data. Lines 18-19 train the classification model and compute the accuracy. Line 20 estimates the $z$-th cumulative reward. Lines 21-22 compute the gradients and update policy functions.

---

**Algorithm 1** Learning policy functions in RMM

1: **Input:** Data $\mathcal{D}$ for generating pseudo CIL tasks.
2: **Output:** Policy functions $\pi_\eta, \pi_\phi$.
3: Initialize $\eta$ and $\phi$;
4: **for** $m$ epochs **do**
5:   **for** $k$ **in** $1, ..., K$ **do**
6:     Create a new pseudo task $\mathcal{T}_k$ using $\mathcal{D}$;
7:     **for** $z$ **in** $1, ..., Z$ **do**
8:       Initialize classification model $\Theta_0$;
9:       **for** $i$ **in** $0, ..., N$ **do**
10:         **if** $i \geq 1$ **do**
11:           Observe $s_i$ and produce $a_i^{[1]} \sim \pi_\eta(s_i)$;
12:           Allocate $\mathcal{M}_{\text{old}}$ and $\mathcal{M}_{\text{new}}$ using Eq. 1;
13:           Produce $a_i^{[2]} \sim \pi_\phi(a_i^{[1]}, s_i)$;
14:           Allocate $\{\mathcal{M}_j^A\}_{j=0}^i$ and $\{\mathcal{M}_j^B\}_{j=0}^i$ using Eq. 2;
15:           Update $\mathcal{E}_{0:i-1}$ using herding [40];
16:           Save $\mathcal{E}_{0:i-1}$ in $\mathcal{M}_{\text{old}}$ and free $\mathcal{M}_{\text{new}}$;
17:         Observe new data and load $\mathcal{D}_i$ into $\mathcal{M}_{\text{new}}$ randomly;
18:         Initialize $\Theta_i$ with $\Theta_{i-1}$ and train it using $\mathcal{E}_{0:i-1} \cup \mathcal{D}_i$;
19:         Compute validation accuracy $r_i$;
20:       Compute $R_z^k = \sum_{i=0}^N r_i$ and update $b$;
21:     Compute $\nabla_{\eta,\phi} J(\eta, \phi)$ using Eq. 5;
22:   Update $\eta$ and $\phi$ using Eq. 6.

---

## 5 Experiments

We evaluate the proposed RMM method on three CIL benchmarks: CIFAR-100 [24], ImageNet-Subset [40], and ImageNet-Full [44], and use two top performing methods LUCIR+AANets and POD+AANets [30] as baselines. Below we introduce the datasets and implementation details (Section 5.1), followed by the experimental results and analyses (Section 5.2).

### 5.1 Datasets and Implementation Details

**Datasets.** We use three benchmarks based on two datasets, CIFAR-100 [24] and ImageNet [44], following common settings [13, 18, 40, 30]. CIFAR-100 [24] contains $60,000$ samples of $32 \times 32$ color images from 100 classes. There are 500 training and 100 test samples for each class. ImageNet (ILSVRC 2012) [44] contains around 1.3 million samples of $224 \times 224$ color images from $1,000$ classes. There are about $1,300$ training and 50 test samples for each class. ImageNet has two CIL settings: ImageNet-Subset is based on a subset of 100 classes; and ImageNet-Full uses the full set of $1,000$ classes. The 100-class data for the ImageNet-Subset are sampled from ImageNet. For the experiments on PODNet [13] and POD-AANets [30], we use the same class orders and hyperparameters as [13]. For the experiments on LUCIR [18] and LUCIR-AANets [30], we use the same class orders and hyperparameters as [18].

**Benchmarks.** We follow the benchmark protocol used in [13, 18, 30, 31]. Given a dataset, the initial (the 0-th phase) model is trained on the data of half of the classes. Then, it learns the remaining classes evenly in the subsequent $N$ phases. Assume there is an initial phase and $N$ incremental phases in the CIL system. The total number of incremental phases $N$ is set to be 5, 10 or 25 (for each the setting is called "$N$-phase" setting). At the end of each individual phase, the learned model in each phase is evaluated on the test set containing all seen classes. In the tables, we report average accuracy over all phases and the last-phase accuracy, where the latter indicates the degree of forgetting.

**Network Architectures.** Following [18, 30, 40, 52], we use a 32-layer ResNet [40] for CIFAR-100 and an 18-layer ResNet [17] for ImageNet. Please note that it is standard to use a shallower ResNet for ImageNet. The 32-layer ResNet consists of an initial convolution layer and three residual blocks (in a single branch). Each block has ten convolution layers with $3 \times 3$ kernels. The number of filters starts from 16 and is doubled every next block. After these three blocks, there is an average-pooling layer to compress the output feature maps to a feature embedding. The 18-layer ResNet follows the standard settings in [17]. We deploy AANets using the same parameters as its original paper [30].

| Method | CIFAR-100 | | | ImageNet-Subset | | | ImageNet-Full | | |
|---|---|---|---|---|---|---|---|---|---|
| | $N$=5 | 10 | 25 | 5 | 10 | 25 | 5 | 10 | 25 |
| LwF [29] | 56.79 | 53.05 | 50.44 | 58.83 | 53.60 | 50.16 | 52.00 | 47.87 | 47.49 |
| iCaRL [40] | 60.48 | 56.04 | 52.07 | 67.33 | 62.42 | 57.04 | 50.57 | 48.27 | 49.44 |
| LUCIR [18] | 63.34 | 62.47 | 59.69 | 71.21 | 68.21 | 64.15 | 65.16 | 62.34 | 57.37 |
| Mnemonics [31] | 64.59 | 62.59 | 61.02 | 72.60 | 71.66 | 70.52 | 65.40 | 64.02 | 62.05 |
| PODNet [13] | 64.60 | 63.13 | 61.96 | 76.45 | 74.66 | 70.15 | 66.80 | 64.89 | 60.28 |
| LUCIR-AANets [30] | 66.88 | 65.53 | 63.92 | 72.80 | 69.71 | 68.07 | 65.31 | 62.99 | 61.21 |
| *w/* RMM (ours) | 68.42 | 67.17 | 64.56 | 73.58 | 72.83 | 72.30 | 65.81 | 64.10 | 62.23 |
| POD-AANets [30] | 66.61 | 64.61 | 62.63 | 77.36 | 75.83 | 72.18 | 67.97 | 65.03 | 62.03 |
| *w/* RMM (ours) | **68.86** | **67.61** | **66.21** | **79.52** | **78.47** | **76.54** | **69.21** | **67.45** | **63.93** |

Table 1: Average accuracies (%) across all phases using two state-of-the-art methods (LU-CIR+AANets and POD+AANets [30]) *w/* and *w/o* our RMM plugged in. The upper block is for recent CIL methods. For fair comparison, we re-implement these methods using our strict memory budget (see "**Memory Budget**" in Section 5.1) based on the public code. The results of using another common budget setting and the detailed numbers (confidence intervals and last-phase accuracies) are provided in the supplementary materials.

For policy functions $\pi_\eta$ and $\pi_\phi$, we use two-layer FC networks. All actions are discretized at $0.1$ intervals to reduce the search space and get a tolerable training overhead.

**Hyperparameters and Configuration.** The training of the classification model $\Theta$ exactly follows the uniform setting in [13, 18, 30, 31]. On CIFAR-100 (ImageNet-Subset/Full), we train it for 160 (90) epochs in each phase, and divide the learning rate by 10 after 80 (30) and then after 120 (60) epochs. Then, we fine-tune the model for 20 epochs using only exemplars (including the preserved exemplars of the new data to be used in future phases). We use an SGD optimizer and an ADAM optimizer for the classification model and policy functions, respectively. More details are given in the supplementary.

**Memory Budget.** There are two popular settings about memory budget in related work. One uses a bounded memory budget with a fixed capacity for all phases [18, 31, 40]. Another one allows the memory budget to grow along with phases [18, 20, 49]. The first one is more strict and thus used as the major setting in our paper (note that the results and analyses using the second setting are given in the supplementary materials). In every benchmark, the total budget of memory depends on the phase number $N$. For example, on CIFAR-100, the total memory budget is set as $7,000$ samples when $N$=5 ($7,000$ samples = 10 classes/phase $\times$ 500 samples/class + $2,000$ samples). Please note that $2,000$ is a bounded memory budget allocated since the 0-th phase for saving exemplars. More clarifications about memory budget are given in the supplementary. For fair comparison, we re-implement related methods and report the results in Table 1 if their original results (in the respective papers) were obtained in a different setting of memory budget.

## 5.2 Results and Analyses

Table 1 presents the results of two state-of-the-art methods (LUCIR+AANets and POD+AANets [30]) *w/* and *w/o* our RMM plugged in, and some recent CIL work [13, 18, 29, 31, 40]. Table 2 shows the ablation study in 6 settings. Figure 4 plots the changes of the average number of exemplars per old/new class for the incremental phases.

**Comparing to the State-of-the-Art.** From Table 1, we make the following observations. 1) Our RMM consistently improves the two top baselines LUCIR+AANets and POD+AANets [30] in all settings. E.g., LUCIR-AANets *w/* RMM and POD-AANets *w/* RMM respectively get 2.7% and 3.1% average improvements on the ImageNet-Subset. 2) Our POD-AANets *w/* RMM achieves the best performances. Interestingly, we find that our RMM can boost performance more when the number of phases is larger. For example, when $N$=25, RMM improves POD-AANets by 3.6% and 4.4% on CIFAR-100 and ImageNet-Subset, respectively. These two numbers are 2.3% and 2.1% when $N$=5.

| | CIFAR-100 | | | | | | ImagNet-Subset | | | | | |
|---|---|---|---|---|---|---|---|---|---|---|---|---|
| Ablation Setting | N=5 | | 10 | | 25 | | 5 | | 10 | | 25 | |
| | Avg | Last | Avg | Last | Avg | Last | Avg | Last | Avg | Last | Avg | Last |
| 1 BaseRow | 66.61 | 57.81 | 64.61 | 55.70 | 62.63 | 52.53 | 77.36 | 70.02 | 75.83 | 68.97 | 72.18 | 63.89 |
| 2 One-level RL | 67.92 | 58.61 | 66.94 | 58.31 | 65.95 | 56.44 | 78.50 | 72.00 | 78.15 | 71.00 | 75.47 | 67.47 |
| 3 Two-level RL (Used) | 68.86 | 59.00 | 67.61 | 59.03 | 66.21 | 56.50 | 79.52 | 73.80 | 78.47 | 71.40 | 76.54 | 68.84 |
| *margin* | +2.3 | +1.2 | +3 | +3.3 | +3.6 | +4 | +2.1 | +3.8 | +2.6 | +2.4 | +4.4 | +5 |
| 4 Two-level RL (T.P.) | 68.62 | 59.40 | 67.22 | 58.20 | 65.82 | 56.20 | 78.81 | 72.42 | 77.68 | 70.77 | 75.29 | 68.81 |
| *margin* | +2 | +1.6 | +2.6 | +2.5 | +3.2 | +3.7 | +1.5 | +2.4 | +1.9 | +1.8 | +3.1 | +4.9 |
| 5 UpperBound RL | 70.00 | 61.12 | 68.36 | 60.00 | 66.56 | 56.74 | 80.01 | 74.31 | 78.95 | 71.97 | 76.99 | 69.14 |
| 6 CrossVal Fixed | 67.50 | 58.48 | 66.69 | 57.19 | 65.73 | 55.51 | 77.96 | 70.31 | 76.70 | 69.08 | 74.18 | 66.10 |

Table 2: The evaluation results in the ablation study (%). "T.P." denotes our results using the **P**olicy functions **T**ransferred from another dataset. "Avg", "Last", and "Used" denote the average accuracy over all phases, the last-phase accuracy, and the results used as ours in Table 1, respectively. BaseRow is from the sota method POD-AANets [30]. Row 2 is for learning Level-1 policy. Row 3 is for learning Level-1 and Level-2 policies in a hierarchical way. Row 4 is for using Transferred Policies (from the other dataset in the table), when RL is costly or impossible on target CIL tasks. The bottom lines are two oracles: training the RL model on the target CIL task (Row 5) and using cross-validation to find the best fixed memory allocation between old and new classes (Row 6).

This indicates that the superiority of our RMM is more obvious in challenging settings (where the forgetting problem is more serious due to the more frequent model re-training through phases).

**Ablation Settings.** Table 2 shows the results of our ablation study. Row 1 is for the baseline method POD-AANets [30]. Row 2 is for learning only Level-1 policy $\pi_\eta$ (where each class gets an even split of the memory). Row 3 is for learning both Level-1 policy $\pi_\eta$ and Level-2 policy $\pi_\phi$ in our proposed hierarchical method, and its results are used in Table 1 as "ours". Row 4 is for using Policy functions Transferred from another dataset (T.P.), which means on the target CIL dataset there is no training of RMM. Here, for CIFAR-100, we use the policy functions learned on ImageNet-Subset, and vice versa. On the last two rows, we show two oracle settings. Row 5 is the upper bound that assumes all past and future data are accessible during training RMM on the target CIL dataset. Row 6 is for using cross-validation (i.e., all past, future, and validation data are accessible) to find the best fixed memory split between old and new class data, e.g., $\frac{old}{new} = 0.7$ is chosen and then used in all phases. The details of chosen split rates are given in the supplementary materials.

**Ablation Results.** *Hierarchical*: In Table 2, when comparing Row 2 to Row 1, it is clear that leveraging reinforcement learning yields better results as it can derive adaptive memory allocation between old and new data. Using class-specific memory management further increases the model performance (i.e., comparing Row 3 to Row 2), even though we divide the classes into only two groups. *T.P.* (**T**ransferred **P**olicy functions): Comparing Row 4 to Row 3, we can see that using transferred policy functions (trained on another dataset) achieves comparable performance, and Row 4 does not require any reinforcement learning on the target CIL dataset. *Oracle*: Comparing Row 3 to Row 5, we see that learning RMM on pseudo CIL tasks is comparable to the upper bound case where all training and validate data are accessible, given the fact that the latter needs higher computational overhead and violates the standard CIL protocol. Row 6 results are consistently lower than ours in Row 3, although cross-validation has access to all past, future, and validation data.

**Allocated Memory.** Figure 4 shows the change of the average number of samples per class in three ablative settings. Solid and dashed lines represent old and new classes, respectively. From the plots, we have two observations. 1) Learning RMM on the pseudo or target CIL tasks (green and orange lines), we can obtain similar memory management results (i.e., actions). This means the learned policy is transferrable in non-stationary continuous environments. This matches the conclusion of continuous adaptation in [2]. 2) Using our RMM method achieved more balanced memory sizes between exemplars and new data. For example, in the 1-st phase of the 5-phase setting, "UpperBound

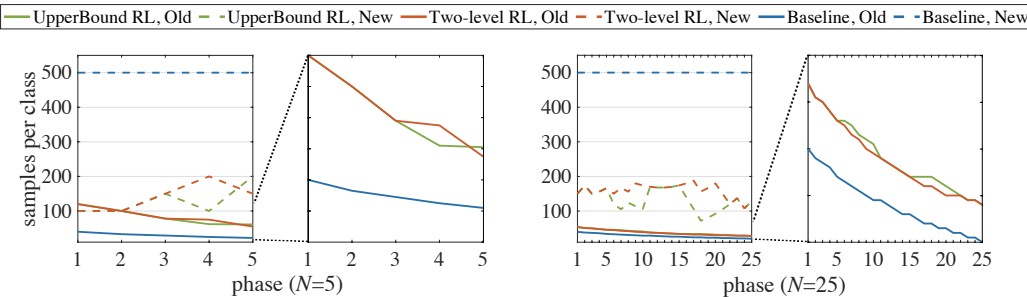

Figure 4: The memory allocated for "Old" and "New" across different phases on CIFAR-100. The second and fourth plots are enlarged versions of the first and third plots, respectively. Solid and dashed lines denote old and new classes, respectively. The baseline is POD-AANets [30]. "Two-level RL" and "UpperBound RL" correspond to Row 3 and Row 5 in Table 2, respectively.

RL" and "Two-level RL" allocate around 100 samples for both exemplars and new data. While the baseline setting has 40 and 500 samples for them, respectively. It thus addresses the data imbalance problem for CIL in a learnable way.

## 6    Conclusions

We propose the reinforced memory management (RMM) method specially for tackling CIL tasks. The hierarchical reinforcement learning (RL) framework (two levels) in RMM is capable of making more adaptive memory allocation actions than using standard RL (one level). Using the generated pseudo tasks in RMM solves the issue of data incompatibility between CIL and RL. Corresponding experimental results show that the policy trained on these pseudo tasks can be directly applied to target tasks without any computational overhead. Our overall method of RMM is generic, and its trained policy (with or without using an in-domain dataset) can be easily incorporated into exemplar replaying-based CIL methods to boost performance.

## Limitations and Societal Impact

We analyse the limitations and potential negative societal impact in the following three aspects.

- *Complexity.* Training RMM takes an additional time cost. According to Algorithm 1, the cost is $O(mKZ)$ times higher than the time used for the target CIL task. However, the training of RMM policy is offline and can use a different dataset (see Table 2) — RMM pre-learns a robust policy from synthesized pseudo tasks and can be directly applied for memory management in real CIL tasks. The overhead of applying this policy is very little, e.g., 0.63% and 1.12% of the total training time respectively on CIFAR-100 and ImageNet (Subset and Full), taking POD+AANets as the baseline.

- *Technical assumptions.* We build the framework of RMM based on a series of technical assumptions, which might not directly hold for all real-world continual-learning applications. When applying our method to mission-critical problems, particular care is required when modeling the system.

- *Privacy issues.* Keeping the old class exemplars has the issue of data privacy. This calls for future research that explicitly forgets or mitigates the identifiable feature of the data.

## Acknowledgments and Disclosure of Funding

This research was supported by A*STAR under its AME YIRG Grant (Project No. A20E6c0101), Alibaba Innovative Research (AIR) programme, and Max Planck Institute for Informatics.

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
