# RMM: Reinforced Memory Management
# for Class-Incremental Learning
## *Supplementary Materials*

Yaoyao Liu[1]    Bernt Schiele[1]    Qianru Sun[2]

[1]Max Planck Institute for Informatics, Saarland Informatics Campus
[2]School of Computing and Information Systems, Singapore Management University
{yaoyao.liu, schiele}@mpi-inf.mpg.de   qianrusun@smu.edu.sg

These supplementary materials include the additional results for comparing to the state-of-the-art (§A), the experiments with a growing memory budget (§B), more details about memory budgets (§C), more ablation results (§D), additional information (§E), the instruction for the PyTorch code (§F) and the checklist (§G).

## A   Additional Results for Comparing to the State-of-the-Art.

This is supplementary to Section 5.2 "**Comparing to the State-of-the-Art.**" In Table S1, we supplement the last-phase accuracies (%) of RMM and some recent CIL methods [3, 5, 8, 10–12]. In Table S2, we supplement the 95% confidence intervals of POD-AANets *w/* RMM.

For Table S1, we can observe: 1) our RMM consistently improves the two top baselines LU-CIR+AANets and POD+AANets [10] in all settings; 2) our POD-AANets *w/* RMM achieves the best performances.

| Method | CIFAR-100 | | | ImageNet-Subset | | | ImageNet-Full | | |
|---|---|---|---|---|---|---|---|---|---|
| | $N$=5 | 10 | 25 | 5 | 10 | 25 | 5 | 10 | 25 |
| LwF [8] | 45.34 | 41.86 | 41.07 | 44.74 | 39.22 | 37.40 | 39.63 | 38.31 | 39.84 |
| iCaRL [12] | 39.01 | 46.16 | 43.04 | 55.76 | 50.32 | 45.72 | 50.57 | 40.44 | 40.52 |
| LUCIR [5] | 54.47 | 52.98 | 49.37 | 60.88 | 56.66 | 50.10 | 58.34 | 51.70 | 46.83 |
| Mnemonics [11] | 55.49 | 53.08 | 49.71 | 64.06 | 62.36 | 60.02 | 58.23 | 56.28 | 52.80 |
| PODNet [3] | 54.95 | 53.29 | 52.17 | 68.08 | 65.57 | 58.50 | 60.28 | 58.61 | 50.63 |
| LUCIR-AANets [10] | 58.75 | 56.88 | 53.89 | 64.52 | 58.90 | 56.73 | 58.32 | 55.69 | 52.78 |
| *w/* RMM (ours) | **60.48** | 58.96 | 54.51 | 67.42 | 66.60 | 63.12 | 58.86 | 55.97 | 53.41 |
| POD-AANets [10] | 57.81 | 55.70 | 52.53 | 70.02 | 68.97 | 63.89 | 61.10 | 57.70 | 53.41 |
| *w/* RMM (ours) | 59.00 | **59.03** | **56.50** | **73.80** | **71.40** | **68.84** | **62.50** | **60.10** | **55.50** |

Table S1: Supplementary to Table 1. Last-phase accuracies (%) using two state-of-the-art methods (LUCIR+AANets and POD+AANets [10]) *w/* and *w/o* our RMM plugged in. The upper block is for recent CIL methods. For fair comparison, we re-implement these methods using our strict memory budget (see "**Memory Budget**" in Section 5.1) based on the public code.

35th Conference on Neural Information Processing Systems (NeurIPS 2021).

| Method | CIFAR-100 | | | ImageNet-Subset | | |
|---|---|---|---|---|---|---|
| | $N$=5 | 10 | 25 | 5 | 10 | 25 |
| BiC [15] | $54.90_{\pm0.38}$ | $51.82_{\pm0.47}$ | $49.72_{\pm0.52}$ | – | – | – |
| LUCIR [5] | $63.34_{\pm0.30}$ | $62.47_{\pm0.29}$ | $59.69_{\pm0.28}$ | $71.21_{\pm0.45}$ | $68.21_{\pm0.41}$ | $64.15_{\pm0.32}$ |
| PODNet [3] | $64.60_{\pm0.15}$ | $63.13_{\pm0.42}$ | $61.96_{\pm0.25}$ | $76.45_{\pm0.40}$ | $74.66_{\pm0.32}$ | $70.15_{\pm0.41}$ |
| POD-AANets [10] | $66.61_{\pm0.91}$ | $64.61_{\pm1.13}$ | $62.63_{\pm0.98}$ | $77.36_{\pm0.06}$ | $75.83_{\pm0.05}$ | $72.18_{\pm0.07}$ |
| POD-AANets [10] *w/* RMM (ours) | $\mathbf{68.86}_{\pm1.02}$ | $\mathbf{67.61}_{\pm0.69}$ | $\mathbf{66.21}_{\pm0.62}$ | $\mathbf{79.52}_{\pm0.05}$ | $\mathbf{78.47}_{\pm0.04}$ | $\mathbf{76.54}_{\pm0.06}$ |

Table S2: Supplementary to Table 1. Average accuracies (%) with 95% confidence intervals.

## B  Experiments with a Growing Memory Budget

This is supplementary to Section 5.2 "**Comparing to the State-of-the-Art.**" In Table S3, we supplement the average accuracies (%) of RMM and some recent CIL methods [3, 5, 6, 8, 10–12, 14, 15] using the setting that allows the memory budget to grow along with phases (i.e., the second setting in Section 5.1 "**Memory Budget**"). In this setting, for saving exemplars, we allocate a memory budget of $1,000$ samples since the 0-th phase, and we allocate an additional memory budget of 20 samples for each new class in each phase. For example, on CIFAR-100 ($N$=5), the total memory budget of the 1-st phase is set as $6,000$ samples ($6,000$ samples = 10 classes/phase $\times$ $500$ samples/class + $1,000$ samples), and the total memory budget of the 2-nd phase is set as $6,200$ samples ($6,200$ samples = 10 classes/phase $\times$ $500$ samples/class + $1,000$ samples + 20 samples/class $\times$ 10 classes).

From Table S3, we can observe: our POD-AANets *w/* RMM achieves the best performances using the setting that allows the memory budget to grow along with phases.

| Method | CIFAR-100 | | | ImageNet-Subset | | |
|---|---|---|---|---|---|---|
| | $N$=5 | 10 | 25 | 5 | 10 | 25 |
| LwF [8] | 49.59 | 46.98 | 45.51 | 53.62 | 47.64 | 44.32 |
| iCaRL [12] | 57.12 | 52.66 | 48.22 | 65.44 | 59.88 | 52.97 |
| BiC [15] | 59.36 | 54.20 | 50.00 | 70.07 | 64.96 | 57.73 |
| LUCIR [5] | 63.17 | 60.14 | 57.54 | 70.84 | 68.32 | 61.44 |
| TPCIL [14] | 65.34 | 63.58 | – | 76.27 | 74.81 | – |
| DDE [6] | 65.42 | 64.12 | – | 76.71 | 75.41 | – |
| Mnemonics [11] | 63.34 | 62.28 | 60.96 | 72.58 | 71.37 | 69.74 |
| PODNet [3] | 64.83 | 63.19 | 60.72 | 75.54 | 74.33 | 68.31 |
| LUCIR-AANets [10] | 66.74 | 65.29 | 63.50 | 72.55 | 69.22 | 67.60 |
| POD-AANets [10] | 66.31 | 64.31 | 62.31 | 76.96 | 75.58 | 71.78 |
| *w/* RMM (ours) | **68.36** | **66.67** | **64.12** | **79.50** | **78.11** | **75.01** |

Table S3: Supplementary to Table 1. Average accuracies (%) across all phases using the growing memory budget. For the related methods [3, 5, 6, 8, 10–12, 14, 15], we directly use the results reported in their original papers.

## C  More Details about Memory Budgets

This is supplementary to to Section 5.1 "**Memory Budget**." In Table S4, we supplement the total memory budgets and the memory budgets when we use cross-validation (i.e., all past, future, and validation data are accessible) to find the best fixed memory split between old and new class data. In every benchmark, the total budget of memory depends on the phase number $N$. For example, on CIFAR-100, the total memory budget is set as $7,000$ samples when $N$=5 ($7,000$ samples = 10

classes/phase $\times$ 500 samples/class + 2,000 samples). Please note that 2,000 is a bounded memory budget allocated since the 0-th phase for saving exemplars.

| Dataset | CIFAR-100 | | | ImageNet-Subset | | | ImageNet-Full | | |
|---|---|---|---|---|---|---|---|---|---|
| | $N$=5 | 10 | 25 | 5 | 10 | 25 | 5 | 10 | 25 |
| Total Memory Budget | 7,000 | 4,500 | 3,000 | 15,000 | 8,500 | 4,600 | 150,000 | 85,000 | 46,000 |
| CrossVal Fixed - New | 1,000 | 500 | 200 | 3,900 | 1,950 | 780 | 52,000 | 26,000 | 7,800 |
| CrossVal Fixed - Old | 6,000 | 4,000 | 2,800 | 11,100 | 6,550 | 3,820 | 98,000 | 59,000 | 38,200 |

Table S4: Supplementary to Section 5.1 "**Memory Budget**." The memory budgets (i.e., the numbers of samples) for different settings. The first block shows the total memory budget, and the second block shows the memory budgets when we use cross-validation (i.e., all past, future, and validation data are accessible) to find the best fixed memory split between old and new class data.

## D   More Ablation Results

**Ablation Results in Unknown Scenarios.** This is supplementary to Section 5.2 "**Ablation Results**." We may not know the number of incremental phases or the classes in future phases in real-world application scenarios. To evaluate the performance of our RMM in unknown scenarios, we supplemented the experiments of using the policy functions trained "in distinct numbers of phases" and "on different datasets" and show the testing results of CIFAR-100 in Table S5. It is clear in the table that even if the policy is learned in a different setting, it does not hurt much compared to the best performance achieved in the same setting (**bold**). For example, when tested on the target CIL task of "CIFAR-100, $N$=25", using the policy learned on "ImageNet-Subset, $N$=5" (65.40%) is only 0.8 percentage points lower than using the policy learned on "CIFAR-100, $N$=25" (66.21%, optimal), and it is 2.8 percentage points higher than the baseline (62.63%). The reason is that our RMM policy is learned on the pseudo CIL tasks synthesized in different settings, i.e., trained in a setting-agnostic manner.

| No. | Method | Policy learned on | Tested on "CIFAR-100 $N$=5" | Tested on "CIFAR-100' $N$=10" | Tested on "CIFAR-100 $N$=25" |
|---|---|---|---|---|---|
| 1 | Baseline | - | 66.61 | 64.61 | 62.63 |
| 2 | | "CIFAR-100, $N$=5" | **68.86** | 67.53 | 65.70 |
| 3 | w/ RMM | "CIFAR-100, $N$=10" | 68.47 | **67.61** | 65.98 |
| 4 | | "CIFAR-100, $N$=25" | 68.28 | 67.06 | **66.21** |
| 5 | | "ImageNet-Subset, $N$=5" | 68.62 | 67.45 | 65.40 |
| 6 | w/ RMM | "ImageNet-Subset, $N$=10" | 68.84 | 67.22 | 65.77 |
| 7 | | "ImageNet-Subset, $N$=25" | 67.96 | 67.04 | 65.82 |

Table S5: Supplementary to Table 2. Average recognition accuracy across all phases (%). Row 1 (baseline) is from the sota method POD-AANets [25]. In Rows 2-7, we show the results for training the RMM policy on pseudo CIL tasks synthesized in one setting and evaluating the policy on the target CIL task in another setting. For example, Row 5 is for training the policy on "ImageNet-Subset, $N$=5" and testing it on "CIFAR-100, $N$=5/10/25".

**Ablation Results for Transferability Properties.** This is supplementary to Section 5.2 "**Ablation Results**." Our learning and application of RMM policy functions do not require datasets to have good transferability properties. To verify this, we supplement the experiments using a new dataset — Non-Overlapping ImageNet (NO-ImageNet) by removing any overlapping classes between ImageNet and CIFAR-100 (if several classes in ImageNet are semantically close to a class on CIFAR-100, all

of them will be removed). We learn the RMM policy on NO-ImageNet and then apply it in the CIL tasks of CIFAR-100. In Table S6, we can see the clear improvements, e.g., 2.97 percentage points on $N$=10 comparing "policy learned on NO-ImageNet" (67.58%) to the baseline (64.61%). The reason is that the memory allocation in RMM is realized through the meta-learned hyperparameters (using pseudo CIL tasks), and hyperparameters have been shown easier to be transferred among different tasks or datasets (than network parameters that encode detailed image patterns). We can find some similar conclusions in few-shot learning that meta-learned hyperparameters can be transferred among different few-shot tasks [9]. The experiment results on the out-of-domain datasets (e.g., stylized ImageNet-Subset [4]) are available in Table S7.

| No. | Method | Policy learned on | Tested on "CIFAR-100 $N$=5" | Tested on "CIFAR-100' $N$=10" | Tested on "CIFAR-100 $N$=25" |
|-----|--------|-------------------|------------------------------|-------------------------------|-------------------------------|
| 1 | Baseline | – | 66.61 | 64.61 | 62.63 |
| 2 | *w/* RMM | CIFAR-100 | 68.86 | 67.61 | 66.21 |
| 3 | | NO-ImageNet | 68.59 | 67.58 | 65.23 |

Table S6: Average recognition accuracy across all phases (%). Row 1 (baseline) is from the sota method POD-AANets [10]. In Rows 2 and 3, we show the results for training the RMM policy on pseudo CIL tasks synthesized in one setting and evaluating the policy on the target CIL task in another setting. For example, Row 3 is for training the RMM policy on the Non-Overlapping ImageNet (NO-ImageNet) and testing it on CIFAR-100 with $N$=5, 10, 25 (the number of phases in training is the same as each test case).

| No. | Method | Policy learned on | $N$=5 | $N$=10 | $N$=25 |
|-----|--------|-------------------|-------|--------|--------|
| 1 | Baseline | – | 49.02 | 44.59 | 38.23 |
| 2 | *w/* RMM | ImageNet-Subset | 53.15 | 50.05 | 42.89 |

Table S7: Average recognition accuracy across all phases (%) on stylized ImageNet-Subset [4]. In this dataset, the class order and data splits are the same as ImageNet-Subset [12], and the images are from stylized ImageNet [4]. Row 1 (baseline) is from the sota method POD-AANets [10]. In Row 2, we show the results for training the RMM policy on pseudo CIL tasks synthesized using ImageNet-Subset and evaluating the policy on the target CIL tasks from stylized ImageNet-Subset.

**Ablation Results for Memory Budgets.** This is supplementary to Section 5.2 "**Ablation Results**." We conduct the experiments by increasing the total memory budget from $1,000$ to $4,000$ and report the results in Table S8. We observe that (1) the improvement by RMM is more significant for a smaller memory budget, and (2) our RMM improves the performance by a clear margin when the memory budget is increased a lot, e.g., to $4,000$. Our explanation is that the effectiveness of RMM is because it relieves the problem of data imbalance between old and new classes. If the memory budget is limited (e.g., $1,000$), RMM is clearly helpful. If the memory budget increases to certain levels (e.g., $2,000$ and $4,000$) that the imbalance between old and new classes is still significant, RMM is also helpful. If the memory budget is unlimited, which means it can be used to store all old-class data, there is no imbalance problem anymore and thus no need to adjust the memory allocation.

**Using the Entropy for Splitting the Data in Groups.** This is supplementary to Section 5.2 "**Ablation Results**." The entropy is often used to measure the uncertainty of the data, e.g., for curriculum learning [1] and open set recognition [2]. We used it to split classes into two groups: (1) high-entropy classes that are more uncertain and need more exemplars; and (2) low-entropy classes that are less uncertain and need fewer exemplars. As suggested in the review, we also try prediction scores in Level-2 (Table S9), and see slightly lower performance (than using entropy), while still achieving satisfactory performance.

# E   Additional Information

**Hardware Information.** We run our experiments using GPU workstations as follows,

| No. | Method | Memory budget of exemplars | $N$=5 | $N$=10 | $N$=25 |
|-----|--------|---------------------------|-------|--------|--------|
| 1 | Baseline | 1000 | 64.31 | 60.97 | 58.77 |
| 2 | *w/* RMM (ours) | 1000 | 68.20 | 65.57 | 63.08 |
|   | margin |  | +3.9 | +4.6 | +4.3 |
| 3 | Baseline | 2000 | 66.61 | 64.61 | 62.63 |
| 4 | *w/* RMM (ours) | 2000 | 68.86 | 67.61 | 66.21 |
|   | margin |  | +2.3 | +3 | +3.6 |
| 5 | Baseline | 4000 | 67.86 | 66.87 | 65.74 |
| 6 | *w/* RMM (ours) | 4000 | 70.12 | 69.05 | 67.72 |
|   | margin |  | +2.3 | +2.2 | +2 |

Table S8: Supplementary to Table 2. Average recognition accuracy across all phases (%) on CIFAR-100, $N$=5/10/25. Row 1 (baseline) is from the sota method POD-AANets [10].

| No. | Method | Tested on "$N$=5" | Tested on "$N$=10" | Tested on "$N$=25" |
|-----|--------|-------------------|--------------------|--------------------|
| 1 | Baseline | 66.61 | 64.61 | 62.63 |
| 2 | Using entropy for grouping (ours) | 68.86 | 67.61 | 66.21 |
| 3 | Using prediction scores for grouping | 67.96 | 67.02 | 66.02 |

Table S9: Supplementary to Table 2. Average recognition accuracy across all phases (%) on CIFAR-100. Row 1 (baseline) is from the sota method POD-AANets [10]. In Rows 2 and 3, we show the results for using different metrics to group classes for Level-2 policy. For example, Row 3 shows the results of using prediction score for grouping classes on "CIFAR-100, $N$=5/10/25".

- **CPU**: AMD EPYC 7502P 32-Core Processor
- **GPU**: NVIDIA Quadro RTX 8000, 48 GB GDDR6
- **Memory**: 1024 GiB = 8x 128GiB, DDR4, 3200 MHz, ECC

**Licenses.** The code for the following papers using the MIT License: AANets [10], iCaRL [12], Mnemonics [11], and PODNet [3].

**About the Datasets.** We use two datasets in our paper: CIFAR-100 [7] and ImageNet [13]. The data for both datasets are downloaded from their official websites and allowed to use for non-commercial research and educational purposes.

# F   PyTorch Code

The code is available at `https://class-il.mpi-inf.mpg.de/rmm/`.

**Getting Started.** To run this repository, we kindly advise you to install python 3.6 and PyTorch 1.2.0 with Anaconda. You may download Anaconda and read the installation instruction on the official website (https://www.anaconda.com/download/).

Create a new environment and install PyTorch and torchvision on it:

```
conda create --name rmm-pytorch python=3.6
conda activate rmm-pytorch
conda install pytorch=1.2.0
conda install torchvision -c pytorch
```

Install other requirements:

```
pip install PyYaml scikit-learn matplotlib pandas requests psutil tqdm
    Pillow==6.2.0
```

**Running experiments on CIFAR-100.**

Run the PyTorch code (data will be downloaded automatically):

```
python run_exp_cifar100.py
```

**Running experiments on ImageNet-Subset.**

Put the training data for ImageNet in:

```
./data/imagenet/train
```

Put the validation data for ImageNet in:

```
./data/imagenet/val
```

Run the PyTorch code:

```
python run_exp_imgnet.py
```