# OpenReview forum: "RMM: Reinforced Memory Management for Class-Incremental Learning"
_NeurIPS.cc/2021/Conference — NeurIPS 2021 Poster_

### Official Review · Reviewer_pi5D · 2021-07-14

**Rating:** 7
**Confidence:** 4

**Summary:**

A common approach in class-incremental learning is to retain a fixed set of examples of the old classes to help avoid forgetting. The paper presents an approach to dynamically retain exemplars for the class-incremental learning problem. Reinforcement learning is used for dynamic memory management.

**Ethical Concerns:**

Not applicable.

**Limitations And Societal Impact:**

The limitations are not listed. Societal impact questionairre is referred to as supplementary material, it would have been nice to answer the questions in the main paper.

**Main Review:**

Pros:
+ The idea of using reinforcement learning for memory allocation is interesting. The way in which policy function is learned on pseudo-CIL tasks is novel to my knowledge.
+ A bi-level action space is defined i.e., first deciding the split between old and new classes and then the memory allocated to each specific class.
+ Extensive experiments are provided on three splits and two datasets.
+ The generalizability of the proposed approach is showcased by plugging it in two recent baseline CIL approaches [25+18, 25+13].
+ Experiments with fixed and growing memory budgets are also performed, the proposed approach appears to help in both cases.

Cons/Questions:
- One main issue I see with the current approach is that the policy is learned on pseudo tasks which may have totally different task distributions, intrinsic properties as compared to the future tasks that we are going to encounter. How much the approach is sensitive to such distribution shifts, e.g., can we learn a robust policy on an altogether different distribution or a set of classes that are much different from the ones going to be encountered in future tasks (as it is quite likely in a realistic setting)?
- RL component will result in additional computational costs. I could not find any comparisons on that. It will be useful to compare the training times with recent approaches and also report the overhead caused by policy learning.
- The paper mentions that the policy can be learned with either an in-domain or out-of-domain dataset, there is a single ablation experiment shown to support this. Can authors provide additional experimental results to support this claim, e.g., leveraging a medical-scan dataset, stylized ImageNet or iNaturalist?
- Since Table 1 reports average accuracies, can the authors report confidence intervals or standard deviations over all tasks (for all/available methods)?
- The memory budget in Table S4 seems to have used high storage. Can the authors confirm its in line with the literature? My understanding is that it's the total memory and not necessarily the memory used for storing the exemplars only. If this is the case, it will be clear to provide exemplars only memory statistics as well.
- In figure 1, it will be helpful to annotate the meaning of different colors (i.e., old exemplars, new exemplars etc.).
- The related works section can be improved with related papers:
-- Zhu et al. Prototype Augmentation and Self-Supervision for Incremental Learning. CVPR'21.
-- Wu et al. Incremental Learning via Rate Reduction. CVPR'21.
-- Simon et al. On Learning the Geodesic Path for Incremental Learning. CVPR'21.
-- Iscen et al. Memory-Efficient Incremental Learning Through Feature Adaptation. ECCV'20.
-- Rajasegaran et al. Random path selection for incremental learning. NeurIPS'19.


**Time Spent Reviewing:**

5

---

> ### Author Response · Authors · 2021-08-10
> **Response to R4**
>
> Thank you for your positive comments and highlighting several strengths of the paper. We will address the weaknesses pointed out in the review. We promise to include all additional results and analyses below in our final version.
>
> &nbsp;&nbsp;
>
> #### `R4-Q1` **Sensitivity to distribution shifts?**
> We agree that CIL methods should not be sensitive to the distribution shifts. We clarify that our RMM aims to learn robust policy functions that are transferable among different settings. To evaluate its performance in handling distribution shifts, we supplemented the experiments of using the policy functions trained "in distinct numbers of phases" and "on different datasets" and show the testing results of CIFAR-100 in Table R6. It is clear in the table that even if the policy is learned in a different setting (data distribution), it does not hurt much compared to the best performance achieved in the same setting (**bold**). For example, when tested on the target CIL task of "CIFAR-100, *N*=25", using the policy learned on "ImageNet-Subset, *N*=5" (65.40%) is only 0.8 percentage points lower than using the policy learned on "CIFAR-100, *N*=25" (66.21%, optimal), and it is 2.8 percentage points higher than the baseline (62.63%). The reason is that our RMM policy is learned on the pseudo CIL tasks synthesized in different settings (Lines 215-219 in the main paper), i.e., trained in a setting-agnostic manner.
>
>
> **Table R6: Average recognition accuracy across all phases (%). Row 1 (baseline) is from the sota method POD-AANets [25]. In Rows 2-7, we show the results for training the RMM policy on pseudo CIL tasks synthesized in one setting and evaluating the policy on the target CIL task in another setting. For example, Row 5 is for training the policy on "ImageNet-Subset, *N*=5" and testing it on "CIFAR-100, *N*=5/10/25".**
>
> | No. | Method | Tested on "CIFAR-100, *N*=5" | Tested on "CIFAR-100, *N*=10" | Tested on "CIFAR-100, *N*=25" |
> | -- | -- | -- | -- | -- |
> | 1 | Baseline | 66.61 | 64.61 | 62.63 |
> | 2 | *w/* RMM (policy learned on "CIFAR-100, *N*=5") | **68.86** | 67.53 | 65.70 |
> | 3 | *w/* RMM (policy learned on "CIFAR-100, *N*=10") | 68.47 | **67.61** | 65.98 |
> | 4 | *w/* RMM (policy learned on "CIFAR-100, *N*=25") | 68.28 | 67.06 | **66.21** |
> | 5 | *w/* RMM (policy learned on "ImageNet-Subset, *N*=5") | 68.62 | 67.45 | 65.40 |
> | 6 | *w/* RMM (policy learned on "ImageNet-Subset, *N*=10") | 68.84 | 67.22 | 65.77 |
> | 7 | *w/* RMM (policy learned on "ImageNet-Subset, *N*=25") | 67.96 | 67.04 | 65.82 |
>
> &nbsp;&nbsp;
>
> #### `R4-Q2` **Additional computational costs of RL?**
>
> It is true that using RL requires additional training time. However, this is not that costly. More importantly, its training is offline and can use a different dataset (see Table R1, Table R2, and Table 2 in the main paper) --- RMM pre-learns a robust policy from synthesized pseudo tasks and can be directly applied for memory management in real CIL tasks. The overhead of applying this policy is very little, e.g., 0.63% and 1.12% of the total training time respectively on CIFAR-100 and ImageNet (Subset and Full), taking POD+AANets as the baseline.
>
> We will also compare the training time of recent approaches and report the overhead caused by policy learning in the final version.
>
> &nbsp;&nbsp;
>
> #### `R4-Q3` **More ablation results for out-of-domain datasets?**
> A good suggestion. We build an out-of-domain dataset Non-Overlapping ImageNet (NO-ImageNet) in the following way. From ImageNet, we remove any overlapping classes between ImageNet and CIFAR-100. If several classes in ImageNet are semantically close to a class in CIFAR-100, we remove all of them. We learn the RMM policy on NO-ImageNet and then apply it in the CIL tasks of CIFAR-100. In Table R7, we can see the clear improvements, e.g., three percentage points on *N*=10 comparing "policy learned on NO-ImageNet" (67.58%) to the baseline (64.61%). The reason is that the memory allocation in RMM is realized through the meta-learned hyperparameters (using pseudo CIL tasks), and hyperparameters have been shown easier to be transferred among different tasks or datasets (than network parameters that encode detailed image patterns).
>
> In addition, we will run experiments on the out-of-domain datasets (as suggested in the review), including medical-scan datasets, stylized ImageNet, and iNaturalist, and will provide the results in the final version.
>
> **Table R7: Average recognition accuracy across all phases (%). Row 1 (baseline) is from the sota method POD-AANets [25]. In Rows 2 and 3, we show the results for training the RMM policy on pseudo CIL tasks synthesized in one setting and evaluating the policy on the target CIL task in another setting. For example, Row 3 is for training the RMM policy on the Non-Overlapping ImageNet (NO-ImageNet) and testing it on CIFAR-100 with *N*=5, 10, 25 (the number of phases in training is the same as each test case).**
>
> | No. | Method | Tested on "CIFAR-100, *N*=5" | Tested on "CIFAR-100, *N*=10" | Tested on "CIFAR-100, *N*=25" |
> | -- | -- | -- | -- | -- |
> | 1 | Baseline | 66.61 | 64.61 | 62.63 |
> | 2 | *w/* RMM (policy learned on CIFAR-100) | 68.86 | 67.61 | 66.21 |
> | 3 | *w/* RMM (policy learned on NO-ImageNet) | 68.59 | 67.58 | 65.23 |
>
> &nbsp;&nbsp;
>
> #### `R4-Q4` **Confidence intervals?**
> We provided the confidence intervals of our method (POD-AANets *w/* RMM) in Table S2 (supplementary). We will include the confidence intervals for all methods in Table 1 (main paper) in the final version.
>
> &nbsp;&nbsp;
>
> #### `R4-Q5` **The memory budget in Table S4?**
> Your understanding is correct. The memory budget in Table S4 (supplementary) is the total memory, and it is exactly the same as the related work [26, 34]. We have explained this in the first paragraph of Section C (supplementary).
>
> In our RMM, the memory for the exemplars is dynamic. In Figure 4 (main paper), we show the memory allocated for “Old” and “New” across different phases on CIFAR-100.
>
> &nbsp;&nbsp;
>
> #### `R4-Q6` **The meaning of different colors in Figure 1?**
> In Figure 1 (main paper), we use orange, blue, and green dots to denote the samples observed in the (*i*-1)-th, *i*-th, and (*i*+1)-th phases, respectively. We will update the caption of Figure 1 to clarify this.
>
> &nbsp;&nbsp;
>
> #### `R4-Q7` **Improving related work?**
> Thanks. We will definitely cite and discuss these related papers in our final version.
>
> &nbsp;&nbsp;
>
> #### `R4-Societal impacts`
> We will move the discussions of negative societal impacts to the main paper and will especially discuss computational costs and privacy issues in the final version.
>
> - *Computational costs.* RL-based methods require intensive usage of computing resources, which is not climate-friendly. It calls for future research into proposing more effective RL training strategies that can speed up the training.
>
> - *Privacy issues.* Keeping old class exemplars has the issues of data privacy. This calls for future research that explicitly forgets or mitigates the identifiable feature of the data.

---

> > ### Comment · Reviewer_pi5D · 2021-08-19
> > **Final Comments**
> >
> > I thank the authors for their responses. My main questions have been reasonably answered.
> >
> > It will be nice to include the new results in the revision and improve the related works too.

---

> > > ### Author Response · Authors · 2021-08-21
> > > **Thank you**
> > >
> > > Thanks very much for your reply. We will include the new results and improve related works in the final version.

---

### Official Review · Reviewer_j8oJ · 2021-07-16

**Rating:** 6
**Confidence:** 4

**Summary:**

The paper proposes a RL-based memory management policy for CIL problem. They have two-step formulation of policy, which first determines how to allocate the memory to old and new tasks, then determines how many samples to store per each class within the task. A standard RL framework of policy gradient method is used to optimize their policy. Experimental results how that their RMM can further improve the recent state-of-the-art methods, such as LUCIR or PODNet.

**Ethical Concerns:**

N.A

**Main Review:**

Applying RL for the memory management is novel, and it does seem that the CIL performance improves with the proposed method. However, my major concern is that the policy should be learned with a pseudo-CIL tasks, utilizing the separate or base classes. Namely, one needs to build exact same setting (i.e., with the same number of incremental learning phases, N) as the target CIL setting, and it does not seem to be practical. At least, it would have been nice if the experiments have the results for the mismatched scenarios -- namely, for the case in which the setting of target and pseudo-CIL is different.

Followings are some more comments:

Pro:
   - the problem of memory management for CIL and applying RL is new.
   - Hierarchical formulation of RL is also new.
   - Positive experimental results are shown. (Improvements over the state-of-the-art)

Con & Questions:
   - Major concern is mentioned above. CIL capability should not be confined to the benchmark data setting (Requirement 2) in line 206). Namely, what if the number of phases N becomes 13 or 17, for example? It seems like the whole RL policy should be re-trained, and this is a critical limitation.
   - How does the performance of RMM change with the total memory budget? Maybe, if the total memory budget increases, the effect of RMM would decrease?

Overall, I think the paper is on the borderline. While the setting and methodology is novel, the practicality of the method seems to be limited. I would also see what other reviewers think and finalize my rating. N/A

----
I have read the author's rebuttal and finalized my score to 6. Thanks.


**Time Spent Reviewing:**

1.5 hour

---

> ### Author Response · Authors · 2021-08-10
> **Response to R3**
>
> Thank you for finding our idea new and our experimental results positive. We will address the questions and weaknesses pointed out in the review. We promise to include all additional results and analyses below in our final version.
>
> &nbsp;&nbsp;
>
>
> #### `R3-Q1` **CIL capability should not be confined to the benchmark data setting?**
>
> We agree that CIL methods should not be confined to the benchmark data setting. Here, we clarify that the policy function learned by RMM is transferable among different settings (e.g., distinct numbers of phases). Therefore, it is not necessary to always re-train it when the setting changes. To verify this, we supplemented the experiments of using the policy functions trained "in distinct numbers of phases" and "on different datasets" and show the results on CIFAR-100 in Table R4. It is clear in the table that even if the policy is learned in a different setting, it does not hurt much compared to the best performance (which uses the policy learned in the same setting). For example, when tested on the target CIL task of "CIFAR-100, *N*=25", using the policy learned on "ImageNet-Subset, *N*=5" (65.40%) is only 0.8 percentage points lower than using the policy learned on "CIFAR-100, N=25" (66.21%), and it is 2.8 percentage points higher than the baseline (62.63%). The reason is that our RMM policy is learned on the pseudo CIL tasks synthesized in different settings (Lines 215-219 of the main paper), i.e., trained in a setting-agnostic manner. Please kindly check more additional experiments in our response to `R1-Q3.2-Part 2` "Requiring the transferability between datasets?".
>
>
> **Table R4: Average recognition accuracy across all phases (%). Row 1 (baseline) is from the sota method POD-AANets [25]. In Rows 2-7, we show the results for training the RMM policy on pseudo CIL tasks synthesized in one setting and evaluating the policy on the target CIL task in another setting. For example, Row 5 is for training the policy on "ImageNet-Subset, *N*=5" and testing it on "CIFAR-100, *N*=5/10/25".**
>
> | No. | Method | Tested on "CIFAR-100, *N*=5" | Tested on "CIFAR-100, *N*=10" | Tested on "CIFAR-100, *N*=25" |
> | -- | -- | -- | -- | -- |
> | 1 | Baseline | 66.61 | 64.61 | 62.63 |
> | 2 | *w/* RMM (policy learned on "CIFAR-100, *N*=5") | **68.86** | 67.53 | 65.70 |
> | 3 | *w/* RMM (policy learned on "CIFAR-100, *N*=10") | 68.47 | **67.61** | 65.98 |
> | 4 | *w/* RMM (policy learned on "CIFAR-100, *N*=25") | 68.28 | 67.06 | **66.21** |
> | 5 | *w/* RMM (policy learned on "ImageNet-Subset, *N*=5") | 68.62 | 67.45 | 65.40 |
> | 6 | *w/* RMM (policy learned on "ImageNet-Subset, *N*=10") | 68.84 | 67.22 | 65.77 |
> | 7 | *w/* RMM (policy learned on "ImageNet-Subset, *N*=25") | 67.96 | 67.04 | 65.82 |
>
> &nbsp;&nbsp;
>
> #### `R3-Q2` **Ablation results for memory budgets?**
>
> A good question and suggestion. We conduct the experiments by increasing the total memory budget from 1,000 to 4,000 and report the results in Table R5. We observe that (1) the improvement by RMM is more significant for a smaller memory budget, and (2) our RMM improves the performance by a clear margin when the memory budget is increased a lot, e.g., to 4,000. Our explanation is that the effectiveness of RMM is because it relieves the problem of data imbalance between old and new classes. If the memory budget is limited (e.g., 1,000), RMM is clearly helpful. If the memory budget increases to certain levels (e.g., 2,000 and 4,000) that the imbalance between old and new classes is still significant, RMM is also helpful. If the memory budget is unlimited, which means it can be used to store all old-class data, there is no imbalance problem anymore and thus no need to adjust the memory allocation. We will explore this using more settings of memory budget in the final version.
>
> **Table R5: Average recognition accuracy across all phases (%) on CIFAR-100, *N*=5/10/25. Row 1 (baseline) is from the sota method POD-AANets [25].**
>
> | No. | Method | Memory budget of exemplars | *N*=5 | *N*=10 | *N*=25 |
> | -- | -- | -- | -- | -- |  -- |
> | 1 | Baseline | 1000 | 64.31 | 60.97 | 58.77 |
> | 2 | *w/* RMM (ours) | 1000 | 68.20 | 65.57 | 63.08 |
> |   | margin |  | +3.9 | +4.6 | +4.3 |
> | 3 | Baseline | 2000 | 66.61 | 64.61 | 62.63 |
> | 4 | *w/* RMM (ours) | 2000 | 68.86 | 67.61 | 66.21 |
> |   | margin |  | +2.3 | +3 | +3.6 |
> | 5 | Baseline | 4000 | 67.86 | 66.87 | 65.74 |
> | 6 | *w/* RMM (ours) | 4000 | 70.12 | 69.05 | 67.72 |
> |   | margin |  | +2.3 | +2.2 | +2 |

---

> ### Author Response · Authors · 2021-09-01
> **Thank you**
>
> We thank the reviewer for upgrading the rating. We will include the additional results and analyses in the revision to improve the paper.

---

### Official Review · Reviewer_79eo · 2021-07-16

**Rating:** 7
**Confidence:** 3

**Summary:**

This work addresses the problem of memory management in Class Incremental Learning (CLI). In this setup, models store exemplars from old classes and must decide what exemplars to drop to make space to new memories. The authors propose a hierarchical reinforcement learning approach where an agent first chooses how much of the old memory to release to make space for new samples and then it chooses how much of the new space allocate to high-entropy samples and low-entropy samples (they divide new classes in two groups). Experiments show that the the proposed method attains better performance than POD-AANets and LUCIR on CIFAR and ImageNet. In the supplementary material they show that the proposed memory management system also improves the performance of already-existing systems. Additional studies show how policies transfer between datasets, the impact of hierarchical RL, and how old and new memories balance over time.


**Limitations And Societal Impact:**

The authors vaguely discuss limitations and potential negative societal impact in the supplementary material. They could broaden the discussion by including some discussion about the additional computational cost of training a model with their method or the privacy issues when keeping exemplar data.

**Main Review:**

# Overall Review
The proposed method is sound, interesting, and it improves the performance of class-incremental learners on multiple datasets. The authors provide ablations, a detailed description of their algorithm, and code. I have some questions about the level-2 actions and the chosen baselines but for the moment I recommend to accept this work.

## Strengths
* The proposed method is technically sound and improves the performance of CI learners.
* The text is clear and easy to understand, the authors did a good job explaining their method.
* The authors provide an ablation study

## Weaknesses
* In the text you cite other CIL works such as DER [45] that you do not include in your tables. Could you explain why? I suspect it is due to a difference in setups but some explanation would be welcome by me and those who are not experts in the particular field of CIL.
* The use of the entropy for splitting the data in groups is not well-motivated. Could you reference previous work or include some experiment showing that using entropy is better than directly learning to predict a score?

# Detailed comments
## Originality
* The authors apply existing techniques in RL to the memory management problem in CIL but to the best of my knowledge they are the first to do such thing.

## Quality
* The overall technical quality is good. My only concern is that the group separation by entropy is not explained.

## Clarity
* In general, the text is well-written and easy to follow.

## Significance
* The proposed idea is sound and interesting for the research community. The experiments show significant improvement.

## Reproducibility
* The authors did a good job towards reproducibility. They include an Algorithm, the code, the implementation details, and how to execute the code in the supplementary material.




**Time Spent Reviewing:**

5

---

> ### Author Response · Authors · 2021-08-10
> **Response to R2**
>
> Thank you for your positive comments and highlighting several strengths of our paper. We will address the weaknesses pointed out in the review. We promise to include all additional results and analyses below in our final version.
>
> &nbsp;&nbsp;
>
>
> #### `R2-Q1` **Including CIL related works in the table?**
>
> For fair comparison, we re-implemented the methods of related works in Table 1 (main paper), using our strict memory budget (see the caption of Table 1). Because most methods either don't have public code or are implemented on another deep learning framework (e.g., TensorFlow), we had to re-implement them ourselves, and thus we had to focus on some of the most promising methods. We will re-implement more methods such as [20, 42, 45] for the final version and include the results in Table 1.
>
> &nbsp;&nbsp;
>
> #### `R2-Q2` **Using the entropy for splitting the data in groups?**
>
> The entropy is often used to measure the uncertainty of the data, e.g., for curriculum learning [C] and open set recognition [D]. We used it to split classes into two groups: (1) high-entropy classes that are more uncertain and need more exemplars; and (2) low-entropy classes that are less uncertain and need fewer exemplars. As suggested in the review, we also try prediction scores in Level-2 (Table R3), and see slightly lower performance (than using entropy), while still achieving satisfactory performance.
>
> [C] "Curriculum Learning." ICML 2009.
>
> [D] "Reducing Network Agnostophobia." NeurIPS 2018.
>
> **Table R3: Average recognition accuracy across all phases (%) on CIFAR-100. Row 1 (baseline) is from the sota method POD-AANets [25]. In Rows 2 and 3, we show the results for using different metrics to group classes for Level-2 policy. For example, Row 3 shows the results of using prediction score for grouping classes on "CIFAR-100, *N*=5/10/25".**
>
> | No. | Method | Tested on "CIFAR-100, *N*=5" | Tested on "CIFAR-100, *N*=10" | Tested on "CIFAR-100, *N*=25" |
> | -- | -- | -- | -- | -- |
> | 1 | Baseline | 66.61 | 64.61 | 62.63 |
> | 2 | Using entropy for grouping | 68.86 | 67.61 | 66.21 |
> | 3 | Using prediction scores for grouping | 67.96 | 67.02 | 66.02 |
>
> &nbsp;&nbsp;
>
> #### `R2-Societal impacts`
>
> Very good suggestions. We will include the discussion about computational costs and privacy issues in the final version.
>
> - *Computational costs.* RL-based methods require intensive usage of computing resources, which is not climate-friendly. It calls for future research into proposing more effective RL training strategies that can speed up the training.
>
> - *Privacy issues.* Keeping old class exemplars has the issues of data privacy. This calls for future research that explicitly forgets or mitigates the identifiable feature of the data.

---

> > ### Comment · Reviewer_79eo · 2021-09-01
> > **Post-rebuttal response**
> >
> > Dear Authors,
> >
> > Initially, I gave a score of 7 because I think the method is an interesting contribution to the CL literature that could become the first step to introducing efficient memory management methods. My main concerns were about the entropy-based mechanism, the comparison with previous works, and the computational cost, which you have either solved or promised to solve in the camera ready.
> >
> > I tend to agree with the other reviewers on the fact that it is not clear that the proposed method will generalize to unseen distributions but after discussion with them it seems that the strengths of the paper still outweigh the weaknesses so I have decided to keep the 7.

---

> > > ### Author Response · Authors · 2021-09-01
> > > **Thank you**
> > >
> > > Thanks a lot for your reply. We will improve the paper according to your comments, and include new results and analyses in the revision.

---

### Official Review · Reviewer_Y6tA · 2021-07-29

**Rating:** 6
**Confidence:** 3

**Summary:**

The paper proposes a new memory sampling strategy which replaces random/herding. The strategy consists of two parts: first level determining the distribution of #samples of each class in memory and second level determining samples to be selected for every class. This is learnt via RL policy. The sampler is trained by emulating a pseudo-CL problem with the data available in the pretraining phase-- where there are an equal number of tasks made as expected in the CL setup to enable transfer. Given a sampling configuration, a CL method trains the model and the performance on the pseudo-CL test set is given as reward to the RL algorithm. This trains the policy components, which optimize sample size and samples per class and train CL. Then, the best policy is used to create the memory which is used to train the actual CL problem with state-of-the-art continual learning approaches. This memory selection achieves better performance than previously used memories across CIFAR100, Imagenet100 and Imagenet1000 datasets.

**Limitations And Societal Impact:**

No substantial negative societal impact that must've been discussed in my view but the section was missing from the draft.

**Main Review:**

2.1) Well written: The paper is very well written and was an enjoyable read. The contributions are clearly stated, acknowledgement is given to the existing body of work to the best of my knowledge. Experiments back motivation the problem as well, demonstrating the need. The paper then proceeds to describe a sufficient solution, results are clear and well-presented.

2.2) Novel idea: The paper introduced a an interesting idea which might seem currently inefficient but can potentially be the first step which can result in nice & efficient future works about memory selection aspects of CL (similar to NASNet {1} for the NAS domain)

2.3) Thorough ablations and supporting experiments: The paper analyzed the proposed approach well ablating each of the components and comparing with a maximum limit (Table 2) and exploring the memory size (Figure 4)/growing memory (supplementary) pre-emptively answering many of my questions.

Weaknesses:

3.1) Potentially violating the CL protocol [Critical]: Since the future is unknown-- continual learning formulation and in real world, formulations don’t have access to how many tasks will arrive in the future. Isn't using this information violating the CL protocol? (and this seems to be critical to the proposed method)

3.2) Complexity and expensiveness don’t justify performance improvement [Major]: The introduced method requires extensive retraining of models to learn the optimal policy function. Furthermore, the memory selection process seems very complex in design for a memory sampler. The accuracy increase is relatively minor, which makes me far less excited about the proposed approach. Another major shortcoming is this requires datasets which have good transferability properties (NAS literature demonstrates that CIFAR and Imagenet happen to have similar properties so transfer works between them, but it's often unlikely)

3.2.1) Why not only use the first level and not second-- it would massively simplify the learning complexity and time while still preserving most performance increases (acc. to Table 2).

Overall:

This seems to be a well-written and explained paper with a novel idea which bridges two domains together and could potentially allow interesting directions to explore. However it seems to me that (i) it violates the CL protocol in a way which is critical to the method’s performance (ii) performance improvements does not justify the complexity introduced: it's very expensive to train and complex in design, the improvement in accuracy is not as great which makes me less excited (if more parameters & information are introduced, improvements in performance are expected). However, I might have misunderstood some of these claims. I request the authors to clarify the points presented in the weaknesses, so that I can consider it in my final rating. I am not familiar with RL literature and hence my feedback is primarily about the CL aspects of the work.

{1} Barret Zoph, Vijay Vasudevan, Jonathon Shlens, Quoc V. Le, Learning Transferable Architectures for Scalable Image Recognition, CVPR18

Post rebuttal:

I am somewhat convinced by the author's rebuttal. I presumed assuming the number of tasks was critical, but it was not. I am quite surprised by the generalizability of the proposed approach.

I suspect that there might be a simpler phenomena happening here aiding the generalizability-- eg: balancing to keep extra samples of older classes in a certain manner or picking certain kind of samples. Discovering what it is might vastly simplify the approach and lead to better understanding of the underlying mechanism. This is potentially nice future direction to build upon, and I encourage the authors to pursue it if possible.

Overall, the method seems too complicated for its performance gains currently. However, I think this provides a method for exploring this space and uncovering better sampling strategies. It is generalizable enough to be valuable tool for the community without violating any critical assumptions. I would upgrade my rating to 6.

**Time Spent Reviewing:**

7

---

> ### Author Response · Authors · 2021-08-10
> **Response to R1**
>
> Thank you for finding our paper well-written, our idea novel, and our experiments thorough. We will address the questions and weaknesses pointed out in the review. We promise to include all additional results and analyses below in our final version.
>
> &nbsp;&nbsp;
>
>
> #### `R1-Q3.1` **Potentially violating the CL protocol?**
>
> We fully agree that we may not know the number of incremental phases or the classes in future phases in real-world application scenarios. To evaluate the performance of our RMM in unknown scenarios, we supplemented the experiments of using the policy functions trained "in distinct numbers of phases" and "on different datasets" and show the testing results of CIFAR-100 in Table R1. It is clear in the table that even if the policy is learned in a different setting, it does not hurt much compared to the best performance achieved in the same setting (**bold**). For example, when tested on the target CIL task of "CIFAR-100, *N*=25", using the policy learned on "ImageNet-Subset, *N*=5" (65.40%) is only 0.8 percentage points lower than using the policy learned on "CIFAR-100, *N*=25" (66.21%, optimal), and it is 2.8 percentage points higher than the baseline (62.63%). The reason is that our RMM policy is learned on the pseudo CIL tasks synthesized in different settings (Lines 215-219 in the main paper), i.e., trained in a setting-agnostic manner.
>
>
> **Table R1: Average recognition accuracy across all phases (%). Row 1 (baseline) is from the sota method POD-AANets [25]. In Rows 2-7, we show the results for training the RMM policy on pseudo CIL tasks synthesized in one setting and evaluating the policy on the target CIL task in another setting. For example, Row 5 is for training the policy on "ImageNet-Subset, *N*=5" and testing it on "CIFAR-100, *N*=5/10/25".**
>
> | No. | Method | Tested on "CIFAR-100, *N*=5" | Tested on "CIFAR-100, *N*=10" | Tested on "CIFAR-100, *N*=25" |
> | -- | -- | -- | -- | -- |
> | 1 | Baseline | 66.61 | 64.61 | 62.63 |
> | 2 | *w/* RMM (policy learned on "CIFAR-100, *N*=5") | **68.86** | 67.53 | 65.70 |
> | 3 | *w/* RMM (policy learned on "CIFAR-100, *N*=10") | 68.47 | **67.61** | 65.98 |
> | 4 | *w/* RMM (policy learned on "CIFAR-100, *N*=25") | 68.28 | 67.06 | **66.21** |
> | 5 | *w/* RMM (policy learned on "ImageNet-Subset, *N*=5") | 68.62 | 67.45 | 65.40 |
> | 6 | *w/* RMM (policy learned on "ImageNet-Subset, *N*=10") | 68.84 | 67.22 | 65.77 |
> | 7 | *w/* RMM (policy learned on "ImageNet-Subset, *N*=25") | 67.96 | 67.04 | 65.82 |
>
> &nbsp;&nbsp;
>
> #### `R1-Q3.2-Part 1` **Complexity and expensiveness?**
>
> It is true that using RL requires additional training time. However, this is not that costly. More importantly, its training is offline and can use a different dataset (see Table R1, Table R2, and Table 2 in the main paper) --- RMM pre-learns a robust policy from synthesized pseudo tasks and can be directly applied for memory management in real CIL tasks. The overhead of applying this policy is very little, e.g., 0.63% and 1.12% of the total training time respectively on CIFAR-100 and ImageNet (Subset and Full), taking POD+AANets as the baseline.
>
> Regarding the accuracy, the improvement brought by our RMM is not minor. We highlight that RMM brings more obvious improvements when the number of phases increases, i.e. when the forgetting problem is more serious. For example, it is shown in Table 2 of the main paper that the improvement of last-phase accuracy (using RMM) is DOUBLED when the number of phases is increased from 5 to 25 (Transferred Policy/T.P. setting).
>
> &nbsp;&nbsp;
>
>
> #### `R1-Q3.2-Part 2` **Requiring the transferability between datasets?**
>
> A good question. Our learning and application of RMM policy functions do not require datasets to have good transferability properties. To verify this, we supplement the experiments using a new dataset --- Non-Overlapping ImageNet (NO-ImageNet) by removing any overlapping classes between ImageNet and CIFAR-100 (if several classes in ImageNet are semantically close to a class n CIFAR-100, all of them will be removed). We learn the RMM policy on NO-ImageNet and then apply it in the CIL tasks of CIFAR-100. In Table R2, we can see the clear improvements, e.g., three percentage points on *N*=10 comparing "policy learned on NO-ImageNet" (67.58%) to the baseline (64.61%). The reason is that the memory allocation in RMM is realized through the meta-learned hyperparameters (using pseudo CIL tasks), and hyperparameters have been shown easier to be transferred among different tasks or datasets (than network parameters that encode detailed image patterns). We can find some similar conclusions in few-shot learning that meta-learned hyperparameters can be transferred among different few-shot tasks [A, B].
>
> [A] "An Ensemble of Epoch-wise Empirical Bayes for Few-shot Learning." ECCV 2020.
>
> [B] "Meta-SGD: Learning to Learn Quickly for Few-shot Learning." arXiv preprint arXiv:1707.09835 (2017).
>
> **Table R2: Average recognition accuracy across all phases (%). Row 1 (baseline) is from the sota method POD-AANets [25]. In Rows 2 and 3, we show the results for training the RMM policy on pseudo CIL tasks synthesized in one setting and evaluating the policy on the target CIL task in another setting. For example, Row 3 is for training the RMM policy on the Non-Overlapping ImageNet (NO-ImageNet) and testing it on CIFAR-100 with *N*=5, 10, 25 (the number of phases in training is the same as each test case).**
>
> | No. | Method | Tested on "CIFAR-100, *N*=5" | Tested on "CIFAR-100, *N*=10" | Tested on "CIFAR-100, *N*=25" |
> | -- | -- | -- | -- | -- |
> | 1 | Baseline | 66.61 | 64.61 | 62.63 |
> | 2 | *w/* RMM (policy learned on CIFAR-100) | 68.86 | 67.61 | 66.21 |
> | 3 | *w/* RMM (policy learned on NO-ImageNet) | 68.59 | 67.58 | 65.23 |
>
> &nbsp;&nbsp;
>
> #### `R1-Q3.2.1` **Why not only using the first level?**
>
> "One-level RL" and "two-level RL" are two variants of our RMM method. "One-level RL" is more computationally efficient (about three times faster on average), and "two-level RL" achieves better model performance (about 0.71 percentage points better on average). There is a trade-off. We offer two options for the convenience of followers.
>
> &nbsp;&nbsp;
>
> #### `R1-Societal impacts`
>
> We mentioned the potential negative societal impacts and limitations in Section E (supplementary). We will move these to the main paper.

---

> ### Author Response · Authors · 2021-08-24
> **Thank you**
>
> We thank the reviewer for accepting our response and appreciate the valuable suggestions and comments (in "post rebuttal")! We will explore more about the underlying mechanism of the method in our future work.

---

### Decision · Program_Chairs · 2021-09-27

**Decision:**

Accept (Poster)

**Comment:**

This paper proposes a reinforcement learning-based for memory management in Class-Incremental Learning (CIL). The reviewers find the paper well written and the idea novel. They also think that the experiments are thorough and positive. There were initially some concerns on the generalizability of the proposed method. The reviewers are generally satisfied by the author responses on the issue. Despite remaining reservations regarding the complexity of the method relative to the performance gains, there is a clear consensus among the reviewers that the paper should be accepted.